# Analysis of Gut Microbial Communities and Functions in *Passer ammodendri* Under Two Extreme Environments

**DOI:** 10.3390/microorganisms13112642

**Published:** 2025-11-20

**Authors:** Yaqi Liu, Peng He, Dongxin Liu, Yang Song, Chenxi Jia, Duochun Wang, Qinghua Jin, Gang Song, Qiang Wei

**Affiliations:** 1School of Medicine, Yanbian University, Yanji 133000, China; yqliu751@163.com (Y.L.); yqinghua@ybu.edu.cn (Q.J.); 2National Pathogen Resource Center, Chinese Center for Disease Control and Prevention (Chinese Academy of Preventive Medicine), Beijing 102206, China; liucommon@126.com; 3National Animal Collection Resource Center, Institute of Zoology, Chinese Academy of Sciences, Beijing 100101, China; hepeng@ioz.ac.cn; 4Division of Infectious Disease, Chinese Center for Disease Control and Prevention (Chinese Academy of Preventive Medicine), Beijing 102206, China; songyang@chinacdc.cn; 5Key Laboratory of Animal Biodiversity Conservation and Integrated Pest Management, Institute of Zoology, Chinese Academy of Sciences, Beijing 100101, China; jiacx@ioz.ac.cn; 6National Institute for Communicable Disease Control and Prevention, Chinese Center for Disease Control and Prevention (China CDC), Beijing 102206, China; wangduochun@icdc.cn

**Keywords:** *Saxaul sparrow* (*Passer ammodendri*), intestinal microbial community, characteristics, functions, extreme environment, full-length 16S rRNA gene sequencing

## Abstract

The gut microbiome regulates multiple physiological processes of the host and plays a significant role in the adaptation of wild animal hosts to extreme environments. The saxaul sparrow (*Passer ammodendri*) is a typical bird species found in the northwest of China, characterized by its strong adaptability to extreme environments. Studying it can help reveal the microbial adaptation mechanisms of the host to extreme environments. Therefore, we conducted a comparative analysis of the intestinal microbial community characteristics and functions of the saxaul sparrow in high-altitude (Pamir Plateau) and desert (Tazhong Town) habitats in Xinjiang. The results of full-length 16S rRNA sequencing and species annotation indicated that the bacterial species composition (relative abundance > 0.1%) of the intestinal microbiota community of the saxaul sparrow was *Candidatus Arthromitus* sp. *SFB rat Yit*, *Escherichia coli*, *Enterococcus faecium*, *Enterococcus faecalis*, and *Klebsiella pneumoniae*, in sequence. In addition, *Lysinibacillus sphaericus* is a unique strain specific to the Tazhong group, while *Stenotrophomonas maltophilia* has a much higher abundance in the Tazhong group than in the Pamir Plateau group. It is worth noting that both groups of samples contain potential opportunistic pathogenic bacteria, such as *Escherichia coli* and *Klebsiella pneumoniae*. The Shannon index of the Pamir Plateau group was lower than that of the Tazhong Town group (*p* = 0.0026), indicating that the intestinal microbial diversity of the Pamir Plateau group was lower than that of the Tazhong Town group. However, there was no significant difference in the ACE index between the two groups and it was not statistically significant (*p* > 0.05). The Beta diversity analysis revealed that the distance between the two groups of samples was considerable (*p* = 0.001), indicating a significant separation. The functional annotation results indicated that the Pamir Plateau group exhibited enhanced capabilities in carbohydrate metabolism, energy metabolism, and DNA damage repair, while the Tazhong Town group demonstrated enhanced lipid metabolism and detoxification abilities. These findings will help reveal the possible impact of the living environment on the composition and function of the intestinal microbiota of the saxaul sparrow, fill the gap in comparative studies of the intestinal microbiota characteristics of the saxaul sparrow in two extreme environments, and provide new theoretical support for subsequent related research.

## 1. Introduction

The gut microbiome is of vital importance to the host, playing an irreplaceable role in regulating physiological processes, including energy metabolism, nutrient absorption, immune defense, and growth and development. In recent years, with the rapid development of high-throughput sequencing technology, the interactions between the intestinal microbial community and the host, as well as the impact of diet, have been extensively studied [1,2] and have gradually become a growing focus in the interdisciplinary field of ecology and microbiology. People are increasingly aware that the living environment can influence the composition of an animal’s gut microbiota [3], playing a role in the process where microorganisms assist the host in adapting to its environment [4]. Studies on mammals have shown that extreme winter conditions can lead to a decrease in the relative abundance of *Firmicutes* in the intestines of white-tailed deer, and an increase in the relative abundance of *Bacteroidetes* [5]. With the increase in altitude, the bacterial diversity and richness in the rumen and feces of Tibetan sheep decreased significantly [6]. The relative abundance of *Firmicutes* and *Lactobacillus* in the intestinal microbiota of five-toed jerbos (*Orientallactaga sibirica*) was significantly reduced, while the relative abundance of *Bacteroidetes* and *Actinomycetes* was significantly increased, which helped five-toed jerboas adapt to drought stress [7]. Similarly to mammals, the gut microbial community structure of birds is susceptible to factors such as geographical environment [8,9] and food resources [10,11,12], which may exhibit significant individual differences. Studies have shown that in the intestinal microbial communities of tree sparrows in high-altitude areas, the abundance of bacteria from *Firmicutes*, namely *Lactobacillus* and *Clostridium sensu stricto 1*, is significantly higher than that in the low-altitude group. These bacteria can provide energy for the host and maintain the stability of the intestinal microecology [4]. Through 16S rRNA sequencing technology, the composition of the intestinal microbial communities in the fecal samples of two high-altitude (approximately 3200 m above sea level) breeding populations of Kentish plover (*Charadrius alexandrinus*) and Tibetan sand plover (*Charadrius altrifrons*) living in the Qinghai-Xizang Plateau was analyzed. It was found that the intestinal microbiota communities of both species were mainly composed of *Firmicutes*, *Proteobacteria*, and *Bacteroidetes*, with *Clostridiales* and *Lactobacillales* as the dominant orders. These microbiota help increase nutrient absorption and metabolic efficiency, and enhance resistance to pathogens [13]. In extremely arid environments, the average relative abundance of *Proteobacteria* is the highest in the intestinal microbiota of cranes, followed by *Firmicutes* and *Bacteroidetes*. At the family level, *Enterobacteriaceae* shows significant enrichment, which helps common cranes adapt to the stress caused by drought. In addition, metabolic pathways such as carbohydrate metabolism and nitrogen metabolism are significantly enriched in extremely dry years [14]. Currently, there are few comparative studies on the differences in gut microbial communities of the same bird species in two extreme environments, and such studies on saxaul sparrows are still lacking.

The saxaul sparrow (*Passer ammodendri*) (Figure 1C,D) is a bird of the *Passeriformes* family endemic to the arid regions of Central Asia, belonging to the resident bird [15], which mainly lives in deserts, semi-deserts, and deserts with sparse shrubs at altitudes below 1000 m, and has strong adaptability to arid environments [16]. It is worth noting that this species is also distributed in the alpine regions of the Pamir Plateau at altitudes exceeding 4500 m. Currently, numerous studies have been conducted on the gut microbiota of birds, including the Eurasian tree sparrow (*Passer montanus*) [17,18] and the house sparrow (*Passer domesticus*) [19,20,21]. However, the relevant research on the saxaul sparrows only involves reproductive behavior [22,23], life history [24], and the development and succession of gut microbes in chicks (annotated to the genus level) [25]. Furthermore, since the birds of the *Passer* can adapt well to urban and other human-dominated environments and coexist with humans, there is a possibility that the pathogens or parasites they carry could spread to humans and livestock, thereby potentially posing a health safety risk [26].

Based on this, this study took the saxaul sparrows in the Pamir Plateau and the hinterland of the Taklamakan Desert (Tazhong Town) as the research object, analyzed the structure of the intestinal microbial community of saxaul sparrows by high-throughput sequencing technology (16S rRNA gene sequencing), and compared the differences in microbiota diversity, dominant bacterial composition and physiological functions (such as energy metabolism, nutrient absorption and other related pathways) of the two populations, in order to reveal the possible influence of the living environment on the differences in the gut microbiome of the saxaul sparrow. The results of this study will provide a scientific basis for elucidating the role of extreme living environments in shaping the bird gut microbiome, offering new theoretical support for the study of bird ecological adaptation in extreme environments.

## 2. Materials and Methods

### 2.1. Fecal Sample Collection

During the period from May to June 2023 and from June to July 2024, a total of 20 fecal samples (n = 20) of the saxaul sparrow (Figure 1C,D) were collected from Tazhong Town, Qiemo County, Bayingoleng Mongololian Autonomous Prefecture, Xinjiang Uygur Autonomous Region (coordinates: 83.66079523° E; 38.97331221° N) (Figure 1B) and the Pamir Plateau (Tashkurgan Tajik Autonomous County, Kashgar Prefecture, Xinjiang Uygur Autonomous Region, average coordinates: 75.361809° E; 37.660574° N) (Figure 1A). Among them, 6 samples were collected from the Pamir Plateau in Xinjiang (n = 6) (the collected samples were labeled as M), and the remaining 14 samples were collected from Tazhong Town (n = 14) (the collected samples were labeled as FKDN). We used fog nets to capture saxaul sparrows in both locations. Due to the limited amount of vegetation cover in Tazhong Town, we chose to catch saxaul sparrows in areas with more vegetation. The captured saxaul sparrows were placed separately into clean bird bags that had been soaked and cleaned in a 10% disinfectant solution until they defecated, and then they were released. After putting on sterile disposable gloves and masks, the samplers use sterile cotton swabs to transfer the excreted fecal samples into sterile cryogenic storage tubes (trying to avoid the surface of the feces). After sampling, the samples were fast frozen in liquid nitrogen then transported to the laboratory using dry ice and placed in the −80 °C ultra-low temperature refrigerator (Thermo Fisher (Suzhou) Instrument Co., Ltd., Suzhou, China) for preservation to conduct subsequent DNA extraction. The investigation and capture of wild animals have been approved by the local forestry authorities, and the processing procedure following bird capture has been reviewed and approved by the Laboratory Animal Ethics Committee of the Institute of Zoology, Chinese Academy of Sciences (Approval ID: IOZ-IACUC-2022-114).

### 2.2. Fecal DNA Extraction, PCR Amplification, and 16S rRNA Gene Sequencing

Following the manufacturer’s instructions, we used the TIANamp Stool DNA Kit (Tiangen Biotech (Beijing) Co., Ltd., Beijing, China) to extract the total DNA of bacteria from stool samples. After extracting the total DNA of the sample, specific primers with barcodes were synthesized based on the full-length primer sequences (F: AGAGTTTGATCCTGGCTCAG, R: GNTACCTTGTTACGACTT). Then, PCR amplification was carried out according to the manufacturer’s experimental operation procedures (SensoQuest GmbH, Göttingen, Germany), and the products were purified, quantified, and normalized to form a sequencing library (SMRT Bell). The constructed library was first subjected to library quality inspection, and the qualified libraries were sequenced using the PacBio Sequel II (Pacific Biosciences of California, Inc., Menlo Park, CA, USA).

### 2.3. Quality Assessment of Sequencing Data

The raw subreads were corrected to generate Circular Consensus Sequences (CCS) (using SMRT Link, version 8.0). Then, the software Lima (version 1.7.0) was used to identify CCS from different samples based on barcode sequences, yielding Raw CCS data. In order to obtain high-quality sequences for subsequent analysis, we first used Trimmomatic (version 0.33) to perform quality filtering on the raw data (with a window size of 50 bp. If the average quality value within the window is lower than 20, the bases at the end of the window will be cut off). The software Cutadapt (version 1.9.1) was used to identify and remove primer sequences (with parameters allowing a maximum mismatch rate of 20% and a minimum coverage of 80% for primer sequence identification). Through length filtering (keeping reads ranging from 1400 to 1600 bp), we eliminated non-target sequences such as short fragment primer dimers and long fragment genomic contamination, obtaining Clean-CCS without primer sequences. Subsequently, USEARCH (version 10) was used to concatenate the paired-end reads (with the minimum overlap length set to 10 bp, the minimum similarity allowed for the overlap region being 90%, and the maximum mismatch base number being 5 bp (default value), and the reads of each sample were concatenated). Finally, UCHIME (version 4.2) was used to identify and remove chimeric sequences (with a similarity > 80%), resulting in high-quality CCS.

### 2.4. Statistical Analysis

#### 2.4.1. OTU Clustering, Taxonomic Annotation, and Phylogenetic Analysis

Using USEARCH (version 10.0), sequences were clustered at a similarity level of 97% (default), and OTUs were filtered using a default threshold of 0.005% of the total number of sequenced sequences. Taxonomic annotation of OTU sequences was performed using a naive Bayesian classifier combined with alignment against the Silva.138 reference database (version 2019-12-16), yielding species classification information for each OTU, which enabled the statistical analysis of community composition at various taxonomic levels (phylum, class, order, family, genus, and species) for each sample. Species abundance tables at different taxonomic levels were generated using QIIME (version 2020.6.0), and community structure diagrams for each taxonomic level were plotted for the samples using R language (version 4.4.1) tools.

#### 2.4.2. Alpha Diversity Analysis

Alpha diversity indices were calculated to reflect the species richness and diversity of the samples. Using QIIME2 (version 2020.6), the Alpha diversity indices of the samples—including the ACE index, Shannon index, and PD whole tree—were evaluated. Subsequently, these indices were visualized as boxplots using the R language (version 4.4.1) to provide an intuitive display of the differences in Alpha diversity among different sample groups. The significance of the differences was tested by default using Student’s *t*-test. In addition, we conducted Shapiro–Wilk normality tests and Levene’s tests on the Shannon index data of the two groups to test for normality and homogeneity of variance, respectively. Since the sample sizes of the two groups were unequal, we used the Mann–Whitney U test to analyze the inter-group differences in the Shannon index.

#### 2.4.3. Beta Diversity Analysis

Beta diversity among groups was assessed using Principal Coordinates Analysis (PCoA), Non-Metric Multidimensional Scaling (NMDS), and UPGMA (Unweighted Pair Group Method with Arithmetic Mean) clustering analysis. Weighted UniFrac distance matrices were calculated and visualized to understand further the structural differences in the gut microbiota of saxaul sparrow between the two regions. Permutational Multivariate Analysis of Variance (PERMANOVA) based on the distance matrix was used to test for significant differences between groups.

#### 2.4.4. Intergroup Significance Analysis

Linear Discriminant Analysis Effect Size (LEfSe) analysis was performed using the Python LEfSe package (version 1.1.1). First, the non-parametric factorial Kruskal–Wallis (KW) sum-rank test was used to identify OTUs with significantly different abundance, and taxa showing significant differences in abundance were identified. The LDA values (default > 4) were then used to estimate the effect size of each taxon’s abundance on the observed differences. Additionally, Metastats analysis was conducted to perform *t*-tests on species abundance data between groups to obtain *p*-values, which were subsequently corrected to yield *q*-values. Finally, species contributing to the compositional differences between the two sample groups were screened based on *p*-values (or *q*-values), with a default threshold of *p* ≥ 0.05.

#### 2.4.5. Functional Gene Prediction Analysis

PICRUSt2 (Phylogenetic Investigation of Communities by Reconstruction of Unobserved States, version 2.3.0) was used to align the feature sequences (16S rRNA) with reference sequences from the Integrated Microbial Genomes (IMG) microbial genome database to construct a phylogenetic tree and identify the “nearest species” for each feature sequence. Based on the known gene content and abundance information of these reference species, the gene information of unknown species was predicted. By integrating pathway information from the KEGG database, the abundance values of various metabolic pathways were determined, thereby predicting the overall functional pathways of the microbial community. BugBase (version 0.1.0) was used to predict the biological-level coverage of functional pathways and biologically interpretable phenotypes within complex microbial communities, including Gram-positive, Gram-negative, mobile element-containing, and oxygen utilization traits. First, the relative abundance of each phenotype was estimated across the full range of coverage thresholds (0 to 1, in increments of 0.01) for each sample. BugBase then selected the coverage threshold with the highest variance across all samples for each OTU. After setting the threshold, BugBase generated a final organism-level phenotype prediction table containing the relative abundance of predicted phenotypes for each sample. Additionally, FAPROTAX (Functional Annotation of Prokaryotic Taxa) was used to perform ecological function annotation and prediction by identifying the genus and species names of bacteria in the two sample groups.

## 3. Results

### 3.1. Analysis of Sequencing Data Quality Evaluation Results

In the number of Raw CCS obtained after splitting the original data based on individual barcode labels (Appendix A), the average number of sequences of individuals in the Pamir Plateau group was 57,480, and the average number of sequences in the Tazhong Town group was 17,308. After identifying and removing primer sequences and performing length filtering to obtain Clean CCS without primer sequences, the average number of sequences in the Pamir Plateau group was 57,458, and that in the Tazhong Town group was 17,308. After the chimeras were identified and removed, the average number of Effective CCS obtained for the Pamir Plateau group was 57,127, and for the Tazhong Town group, it was 16,055. After data preprocessing, a total of 535,221 full-length 16S rRNA gene sequences were obtained from 20 samples (Appendix A). A total of 526 OTUs were clustered. Among them, there were 102 common OTUs between the two groups of samples, 116 OTUs specific to the Pamir Plateau group, and 308 OTUs specific to the Tazhong Town group (Figure 2). The average length of the sequences for each sample ranged from 1427 to 1471 bp (averaging around 1454 bp), meeting the screening criteria for the CCS sequence length threshold (16S: 1200–1650 bp) (Appendix A). After data preprocessing, the Effective (%) (the percentage of Effective CCS to Raw CCS) was above 86% (the average was about 94.76%), and the quality was high (Appendix A). In addition, the library coverage rate for all samples was above 99.45%, indicating that the sequencing depth accurately reflected the actual microorganism population in the samples (Appendix A).

### 3.2. Composition and Relative Abundance of Gut Microbiota

We annotated 526 OTUs to 22 phyla, 32 classes, 73 orders, 129 families, 241 genera and 340 species (Appendix A) through species annotation, thereby obtaining the composition of the intestinal microbial community of the saxaul sparrow: the phyla level (Figure 3A, Appendix A) includes *Firmicutes*, *Proteobacteria*, *Campylobacterota*, *Actinobacteriota* and *Bacteroidota* (unclassified, unassigned and uncultured species are not included in the ranking, those with relative abundance <0.1% are not listed either, the same below); the class level (Figure 3B, Appendix A) includes *Clostridia*, *Gammaproteobacteria*, *Bacilli*, *Campylobacteria*, *Actinobacteria*, and *Alphaproteobacteria*; the order level (Figure 3C, Appendix A) includes *Clostridiales*, *Enterobacterales*, *Lactobacillales*, *Mycoplasmatales*, *Campylobacterales*, and *Burkholderiales;* the family level (Figure 3D, Appendix A) includes *Clostridiaceae*, *Enterobacteriaceae*, *Enterococcaceae*, *Mycoplasmataceae*, and *Comamonadaceae*; the species level (Figure 3F, Appendix A) includes *Candidatus Arthromitus*, *Escherichia*, *Shigella*, *Enterococcus*, *Mycoplasma*, and *Klebsiella*. The dominant species (Figure 3F, Appendix A) are *Candida* species *SFB-rat-Yit*, *Escherichia coli*, *Enterococcus faecium*, *Enterococcus faecalis*, and *Klebsiella pneumoniae*.

However, the relative abundance of dominant bacteria at each taxonomic level differed between the two groups, and the types of bacteria at each classification level in the Tazhong Town group were higher than those in the Pamir Plateau group. Taking the species level as an example (Table 1), only 5 bacterial species with a relative abundance of >0.1% were found in the Pamir Plateau group, while there were 7 such species in the Tazhong Town group. Among them, the *Lysinibacillus sphaericus* did not exist in the Pamir Plateau group, and it was a unique strain in the Tazhong Town group. On the other hand, the abundance of *Stenotrophomonas maltophilia* in the Tazhong Town group was much higher than that in the Pamir Plateau group.

### 3.3. Differences in Richness, Diversity, and Composition Structure of Gut Microbiota

The alpha diversity of the gut microbiome in the two groups was evaluated using the ACE abundance index (Figure 4A) and Shannon diversity index (Figure 4B). The Shannon index of the Pamir Plateau group was lower than that of the Tazhong Town group (*p* = 0.0026) (Shapiro–Wilk normality test: *p* > 0.05 (Pamir Plateau group: *p* = 0.1591, Tazhong Town group: *p* = 0.2592); Levene’s test: *p* > 0.05 (*p* = 0.2665); Mann–Whitney U test: *p* < 0.05 (*p* = 0.0187)), indicating that the intestinal microbial diversity of the Pamir Plateau group was lower than that of the Tazhong Town group. However, there was no significant difference in ACE index between the two groups, and the difference was not statistically significant (*p* > 0.05). The principal coordinates analysis (PCoA) based on the weighted UniFrac distance matrix revealed that the two groups of samples were distinctly separated from each other (Figure 4C). The UPGMA (Unweighted Pair Group Method with Arithmetic Mean) sample clustering tree analysis result (Figure 4D) indicated that each group of samples clustered into a single branch, with branch lengths within the groups being relatively short. Eventually, the two groups of samples merged into one branch with a longer branch length, indicating that the phylogenetic relationship between the intestinal microbiomes of the two groups is relatively distant. PERMANOVA analysis was used to test whether there were significant differences in Beta diversity between the two groups of samples (Figure 4E). The results showed that the differences between the groups were statistically significant (*R*^2^ = 0.843, *p* = 0.001).

The results of LEfSe analysis (Figure 5A, Appendix A) revealed a total of 48 marker species with significant differences between the two groups, of which 7 belonged to the Pamir Plateau group and 41 belonged to the Tazhong Town group. The species that contributed the most to distinguishing the gut microbiomes of the two groups of samples belonged to two kingdoms, three phyla, four classes, eight orders, ten families, ten genera, and eleven species, respectively. A *t*-test was performed on the species abundance data between the groups using Metastats analysis to illustrate further the differences in microbial community abundance between the two groups. The results (Figure 5B, Appendix A) showed that there were 58 strains with highly significant differences (*p* < 0.01, **) and 35 strains with significant differences (0.05 < *p* < 0.01, *).

### 3.4. Functional Prediction of Gut Microbiota in Two Groups of Samples

The feature sequences (16S rRNA gene sequences) were subjected to functional prediction in the KEGG functional database using PICRUSt2. Microbial phenotypic prediction and ecological function prediction were conducted through BugBase and FAPROTAX. The KEGG functional annotation results (Appendix A) show that, in terms of substance and nutrient metabolism, the intestinal microbiota of the Pamir Plateau group samples have stronger functions in galactose metabolism, secondary bile acid biosynthesis, other polysaccharide degradation, and glycosaminoglycan degradation than those of the Tazhong Town group. In addition, the Pamir Plateau group also exhibits stronger capabilities in DNA stability-related processes, such as base excision repair, nucleotide excision repair, and mismatch repair, than the Tazhong Town group. In contrast, the intestinal microbiome of the samples from the Tazhong Town group exhibited significantly stronger functions in various metabolic pathways such as fatty acid elongation, steroid biosynthesis, steroid degradation, primary bile acid biosynthesis, D-arginine and D-ornithine metabolism, N-glycan biosynthesis, biosynthesis of other types of O-glycan, mannosyl O-glycan biosynthesis, gastric acid secretion, protein digestion and absorption, as well as detoxification functions such as degradation of fluorobenzoate, degradation of polycyclic aromatic hydrocarbons, degradation of atrazine, and resistance to platinum-based drugs. These functions were significantly stronger than those of the Pamir Plateau group. In terms of energy metabolism, overall, the energy metabolism of the Pamir Plateau group was stronger than that of the Tazhong Town group. Specifically, the intestinal microbiome of the Pamir Plateau group exhibited stronger functions in oxidative phosphorylation, methane metabolism, carbon fixation in prokaryotes, and nitrogen metabolism compared to the Tazhong Town group. In contrast, the intestinal microbiome of the Tazhong Town group showed stronger functions in sulfur metabolism compared to the Pamir Plateau group.

The results of BugBase phenotypic prediction (Figure 6) showed that the relative abundance of OTU of anaerobic bacteria (*Firmicutes*) in the gut microbiome of the Pamir plateau group was significantly higher than that of the Tazhong Town group (*Spirophytes*) (*p* = 0.04). In comparison, the relative abundance of OTUs of aerobic bacteria (*Proteobacteria*) in the Tazhong Town group was higher (*p* = 0.11), although this difference was not statistically significant. The relative abundance of OTU of Gram-positive bacteria (*Firmicutes*) in the Pamir Plateau group was more than 3 times that of the Tazhong Town group (*p* = 0.02). In contrast, the relative abundance of OTU of Gram-negative bacteria (*Proteobacteria* and *Spirophytes*) in the Tazhong Town group was higher than that of the Pamir Plateau group (*p* = 0.02). In addition, the relative abundance of OTUs in strains containing mobile elements of the Pamir Plateau group was lower than that of the Tazhong Town group (*p* = 0.13), although this difference was not statistically significant.

The FAPROTAX ecological function prediction results (Appendix A) showed that the gut microbiota of the Pamir Plateau group lacked functions such as methanotrophy, dark sulfide oxidation, dark sulfur oxidation, aliphatic non-methane hydrocarbon degradation, or dark iron oxidation. In contrast, the gut microbiota of the Tazhong Town group lacked functions such as sulfate respiration and respiration of sulfur compounds. The intestinal microbiome of the Tazhong Town group has stronger functions in arsenite oxidation detoxification, aromatic compound degradation, dissimilatory arsenite oxidation, nitrate denitrification, nitrite denitrification, nitrous oxide denitrification, and denitrification, as well as chitinolysis, compared to the Pamir Plateau group. On the other hand, the Pamir Plateau group demonstrated greater functional potential in processes such as nitrite ammonification, nitrite respiration, iron respiration, and fumarate respiration compared to the Tazhong Town group.

## 4. Discussion

The Pamir Plateau and Tazhong Town are located in the southwest of Xinjiang and the central part of the Tarim Basin. The geographical environment and ecological conditions of the two places are significantly different: The Pamir Plateau in the southwest of Xinjiang belongs to the eastern part of the Pamir Plateau, consisting of highland mountains and high-altitude intermountain basins, respectively, with an average altitude of 4500 m. The climate is cold and humid, with the lowest temperature reaching −50 °C and the highest temperature only around 20 °C. The annual average temperature is only 3–5 °C, and the average temperature in June and July is about 16 °C. The vegetation is primarily composed of alpine meadows and shrubs [27]. In contrast, Tazhong Town is located in the heart of the Taklimakan Desert and has a typical warm temperate continental arid climate. The highest temperature is around 39 °C, and the lowest temperature is generally below −20 °C. The annual average temperature is 10–12 °C, and the average temperature in May and June is about 21 °C. The annual precipitation is extremely scarce, with an average of no more than 100 mm, the lowest being only 4–5 mm, and the average being less than 20 mm. The surface is mainly composed of mobile sand dunes and sparse desert vegetation (such as saxaul (*Haloxylon ammodendron* (C. A. Mey.) Bunge) and camel thorn (*Alhagi camelorum* Fisch.)) [28]. The Pamir Plateau and the Tazhong Town, as typical representatives of the high-cold regions and arid regions in Xinjiang, the significant differences in environmental gradients (altitude, temperature, humidity) and resource availability may drive the differentiation of the intestinal microbiota of the saxaul sparrow in composition, richness and function, which may further affect the energy metabolism efficiency and nutrient absorption capacity of the host.

Our research indicates that there are significant differences in the composition of the intestinal microbial communities, as well as the diversity and relative abundance of various taxa, between the two habitats of the saxaul sparrows. The house sparrow and the Eurasian tree sparrow, like the saxaul sparrow, are all birds belonging to the *Passer*. Studies have shown that the dominant phyla (with relative abundance >5%) in the intestinal microbial community of house sparrows are the *Campilobacterota* (94%), *Firmicutes* (14%), *Ignavibacteriae* (10.9%), *Proteobacteria* (8.6%), *Latescibacteria* (6.2%), and *Actinobacteria* (6%) [19]. In contrast, in the intestinal microbial community of tree sparrows in high-altitude summers, the dominant phyla are *Firmicutes* (56.62%), *Proteobacteria* (30.12%), and *Campylobacter* (12.35%), and in low-altitude summers, they are *Proteobacteria* (40.24%), *Bacteroidetes* (14.86%), and *Firmicutes* (10.07%) [4]. Our analysis results show that *Firmicutes* and *Proteobacteria* are the two most abundant bacterial phyla in the intestinal microbiome of the saxaul sparrow, which is consistent with previous studies that have demonstrated these two phyla are the primary intestinal bacterial groups in the bird’s intestinal microbiome [9,29,30]. However, there was a significant difference in abundance between the two groups: the *Firmicutes*, which had a significant advantage in the Pamir Plateau group, showed a significant decrease in abundance in the Tazhong Town group, while the *Proteobacteria* showed the opposite trend. Furthermore, the abundances of the *Nanoarchaeota*, *Spirochaetes*, *Actinobacteria,* and *Bacteroidetes* in the Tazhong Town group were significantly higher than those in the Pamir Plateau group. Consistent with the previous research results, the ratio of the *Firmicutes*/*Bacteroidetes* (F/B) in the Pamir Plateau group was significantly higher, which might be related to the conversion of cellulose to short-chain fatty acids, which can better meet the energy requirements and help them survive in the relatively harsh dietary environment of the plateau [31,32,33,34]. Studies have shown that the *Nanoarchaeota* can be found in geothermal environments (such as geothermal springs), indicating that it has the property of high tolerance to high temperatures [35,36]. The relatively high abundance of *Nanoarchaeota* bacteria in the Tazhong Town group may be related to the fact that Tazhong Town is located in the heart of the desert, with higher surface temperatures. Furthermore, the Tazhong Town group also recorded relatively high abundances of *Lysinibacillus sphaericus* and *Stenotrophomonas maltophilia*, while the former was absent in the Pamir Plateau group. Studies have shown that *Lysinibacillus sphaericus* can withstand high temperatures and dry environments due to its ability to form spore structures [37], which is beneficial for the adaptation of saxaul sparrows to the desert arid environment. At the same time, *Stenotrophomonas maltophilia* can utilize simple sugars such as glucose [38], which is in line with the relatively low intake of high-fiber food by the saxaul sparrows in the Tazhong Town group. At the species level, the relative abundance of the *Candidatus Arthromitus* sp. *SFB rat Yit* was higher, and it was mainly concentrated in the Pamir Plateau group, while it was relatively lower in the Tazhong Town group. Studies have shown that a high relative abundance of the *Candidatus Arthromitus* can help the host inhibit the colonization of pathogenic microorganisms such as *Escherichia coli O103* [39] and *Salmonella typhimurium* [40] in the intestinal tract, thereby contributing to maintaining the intestinal microecological balance and health of the host [41]. This might also indicate that the ability of the saxaul sparrows in the Pamir Plateau group to maintain a healthy intestinal state is stronger than that of the Tazhong Town group. There were also significant differences in the Alpha and Beta diversity of the intestinal microbial communities between the two groups. The diversity index of the group from Tazhong Town was relatively higher, indicating that the intestinal microbial community of the saxaul sparrows living near Tazhong Town was more diverse. The results of LEfSe analysis further revealed the significant differences between the samples of different groups. Our functional annotation results show that the galactose metabolism, glycan degradation ability and other carbohydrate metabolic capabilities of the Pamir Plateau group are generally higher than those of the Tazhong Town group, reflecting its adaptability to the high-fiber characteristics of alpine meadow plants. The nitrite ammonification and other pathways help to better detoxify and efficiently utilize the limited nitrogen sources, thereby maintaining the nitrogen balance in the body. Furthermore, the energy metabolism capacity of the Pamir Plateau group is generally stronger than that of the Tazhong Town group. This is conducive to their better adaptation to the high energy demands of the high-altitude environment. The lipid metabolism functions such as fatty acid elongation and steroid metabolism in the Tazhong Town group support its efficient utilization of seed oils and insect proteins. Furthermore, studies have shown that compared to a natural environment, exposure to a noisy environment can increase the diversity of intestinal microorganisms in birds to a certain extent [42], which provides some evidence for the fact that the intestinal microbiome diversity in the Tazhong Town group is more abundant. These differences could, in part, be attributed to the diet: In June and July, the rainfall [43] and temperature [44] on the Pamir Plateau have already been relatively high, and at this time, the seeds of alpine meadow plants and the fruits of low-growing shrubs (such as berries and achene) have begun to mature and are abundant [45]. Many insects (such as rhinoceros beetles of the *Beetle* family, locusts of the *Orthoptera*, and larvae of the *Lepidoptera*, etc.) [46,47,48] have also become active and can serve as food sources for the saxaul sparrows. However, the saxaul sparrows in Tazhong Town can not only consume the fruits of drought-resistant shrubs such as *Haloxylon ammodendron* and *Calligonum mongolicum* Turcz., seeds of *Tamarix tarimensis* and other species [49,50], as well as insects, but also, due to their proximity to human settlements, they can pick up various human food residues for consumption [17]. As a result, their intestinal microbiome has become more diverse and complex. On the other hand, it might be caused by the altitude: Previous studies have shown that as altitude increases, the diversity and richness of the intestinal microbiome in birds decrease [51], which to some extent led to the reduction in the diversity and richness of the intestinal microbiome in the Pamir Plateau group. Furthermore, compared to the relatively simple and pure environment of the Pamir Plateau, the living environment of the saxaul sparrow in the Tazhong Town group is closer to human settlements and has a lower altitude [52]. The toxic substances it comes into contact with will be significantly higher than those of the Pamir Plateau group. Therefore, its detoxification functions, such as platinum drug resistance, polycyclic aromatic hydrocarbon degradation, arsenite oxidation detoxification, and aromatic compound degradation, are stronger, enabling it to cope with the chemical pollution caused by human activities. The relative abundance of bacterial OTUs with mobile elements in the Pamir Plateau group is lower than that in the Tazhong Town group, which may also be a contributing factor. Due to its high altitude, the Pamir Plateau has stronger ultraviolet radiation [53] and lower oxygen content [54] compared to Tazhong Town. Therefore, the group from the Pamir Plateau has a stronger ability related to DNA damage repair than the group from Tazhong Town, which can maintain the DNA stability of the intestinal microbial community of the saxaul sparrow and is more conducive to its adaptation to the high-altitude environment.

Our research also revealed that both groups of saxaul sparrows contained opportunistic bacterial pathogens that could potentially cause diseases in humans, indicating that they both have the potential to serve as vectors for the spread of these pathogens. Specifically, in the Pamir Plateau group, the *Escherichia-Shigella* (25.26%) was the dominant genus, with *Escherichia coli* (25.26%) being the dominant species. The KEGG functional prediction results also indicated a potential link to human diseases through pathogenic *Escherichia coli*. The difference was that in the Tazhong Town group, the *Klebsiella* was the most abundant genus, and *Klebsiella pneumoniae* was the dominant species. However, the KEGG results did not show that it would cause human diseases through *Klebsiella pneumoniae*, which might be because the saxaul sparrows in the Tazhong Town area often picked up human food, which led to adaptive evolution of this bacterium in their intestines, resulting in weakened pathogenicity and virulence, and thus not causing diseases in humans [55]. However, *Klebsiella pneumoniae*, as a pathogenic bacterium listed in the “Catalogue of Pathogenic Microorganisms Infecting Humans”, is classified as a Category 3 pathogen [56]. Therefore, it is still necessary to pay attention to the potential harm it may cause to humans and take appropriate preventive measures.

Through literature search, it was found that there is still a research gap regarding the composition of the intestinal microbial community of adult saxaul sparrows at the species level and the comparison of the composition of the intestinal microbial community of saxaul sparrows under different extreme living environments. Moreover, saxaul sparrows can coexist with humans, and the pathogenic bacteria, parasites, and other pathogens they carry may be transmitted to humans during their contact with humans [26]. Therefore, researching the intestinal microbial community of the saxaul sparrow not only holds significant biological research value but also has considerable public health significance. Furthermore, as one of the very few artificial towns built in the middle of a desert globally, the comparative study conducted by Tazhong Town on the intestinal microbial community diversity of the saxaul sparrows living in its vicinity and the same species of birds on the Pamir Plateau holds significant pioneering significance. Moreover, this study conducted full-length sequencing of the 16S rRNA gene and refined the species annotation level to the species level. Compared with previous studies that sequenced the variable regions V3–V4 [4,17,57], V4 [9,58,59], V3–V5 [1], and V6–V8 [60] and only annotated to the family [61] or genus [59,62] level, this study is more accurate and helps deepen our understanding of the intestinal microbial community of the saxaul sparrow. However, this study also has several limitations. Firstly, compared with previous similar studies [57,63], the sample size in this study (n = 20) is relatively small, which may result in some bacteria not being detected during the species annotation process, leading to false negatives in the results and thereby affecting the validity of the statistical results. Secondly, at the species level, some bacterial strains have not yet been accurately identified, such as unclassified *Archaea* and unclassified *Woesearchaeales*, which may, to some extent, limit our in-depth analysis of the adaptation mechanism of the intestinal microbiome of the saxaul sparrow to high-altitude and arid environments. In addition, our study focused solely on the breeding period (May to July) [16] of the saxaul sparrow but did not cover the seasonal variations throughout the year. Therefore, it was unable to reflect the dynamic adaptation of the microbiome (such as food shortages during the wintering period may significantly alter the bacterial community structure [64]). Finally, our study lacked the systematic quality control and reproducibility assessment of sequencing data, and was unable to evaluate the reproducibility between runs or within samples to confirm data consistency. Therefore, future research needs to address these deficiencies to make the research data more complete and obtain more convincing results.

## 5. Conclusions

Our research has, for the first time, revealed that the gut microbial community of the saxaul sparrow at the species level is composed of *Candidatus Arthromitus* sp. *SFB rat Yit*, *Escherichia coli*, *Enterococcus faecium*, *Enterococcus faecalis*, and *Klebsiella pneumoniae*. By analyzing the differences in food and environment between the two locations, we have preliminarily demonstrated that the living environment may affect the composition, richness, and diversity of the gut microbial community of the saxaul sparrow, thereby possibly causing differences in the sparrows’ absorption and utilization of the nutrients in the food they consume, their resistance to pathogenic bacteria in the body, and the degradation of toxic substances in the body. Our research indicates that the diversity of the gut microbiome may be positively correlated with the diversity of the diet. The environment may enable certain related strains (such as those resistant to cold, drought, and detoxification) to colonize the intestines of saxaul sparrows, thereby helping the host adapt to the environment. However, as discussed earlier, in this study, the specific dietary composition of the saxaul sparrow was not tracked. Much of the information regarding its dietary characteristics was obtained from relevant literature, which may have deviations from the actual situation. Further studies on diet tracking and other related aspects are needed to make the results more reliable. In conclusion, our research results will provide important insights into the microbial adaptation mechanism of the saxaul sparrow in extreme environments and offer new theoretical support for future studies on the ecological adaptation of birds under extreme environments.

## Figures and Tables

**Figure 1 microorganisms-13-02642-f001:**
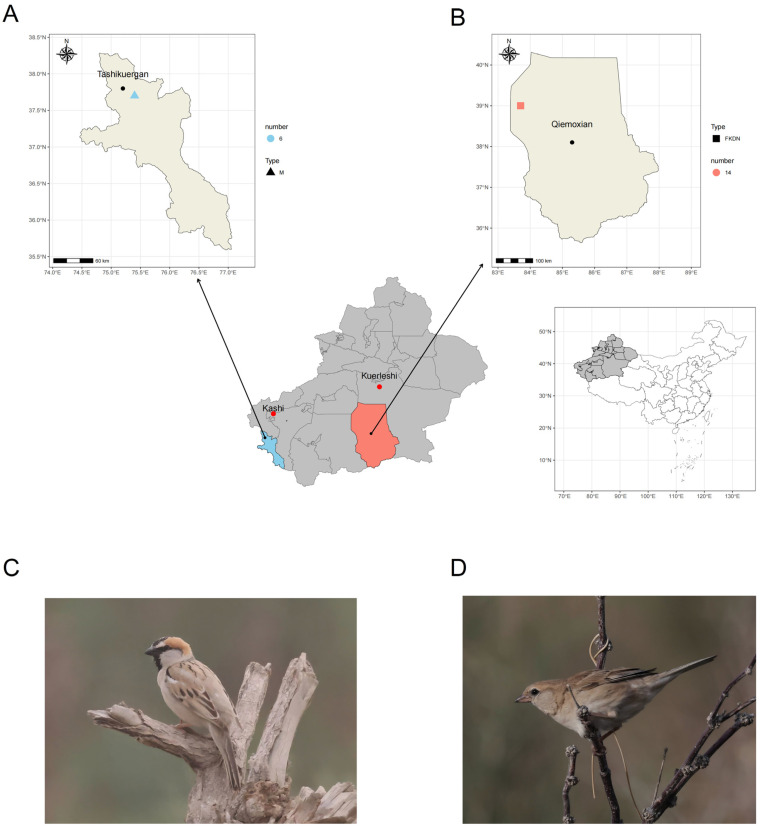
Coordinate map of sampling points for saxaul sparrow fecal samples and photo of the saxaul sparrow. (**A**) Coordinate map of sampling points in the Pamir Plateau group. The blue triangle represents the collection location of the fecal samples of this group of saxaul sparrows. The blue color indicates the quantity of fecal samples collected in the Pamir Plateau (n = 6), and the triangle represents the Pamir Plateau group. The black dot represents the county of Tashkurgan in the Kashgar region. (**B**) Coordinate map of sampling points in the Tazhong Town group. The orange square represents the collection site of the saxaul sparrow feces samples from the Tazhong Town group. The color indicates the quantity of fecal samples collected in Tazhong Town (n = 14). The square represents the Tazhong Town group. The black circle represents the county seat of Bayingol Mongolian Autonomous Prefecture, Qiemo County. (**C**) Photo of a male saxaul sparrow. The male bird has a broad black crest on the center of its head, white eyebrow lines, and its beak, lower jaw and throat are black. (**D**) Photo of a female saxaul sparrow. The female bird has a dark coloration but has dark-brown longitudinal stripes on its upper back and light-colored tips on its middle and large flight feathers. Its beak is yellow with a black tip. The photos were taken and provided by Professor Song Gang from the Institute of Zoology, Chinese Academy of Sciences.

**Figure 2 microorganisms-13-02642-f002:**
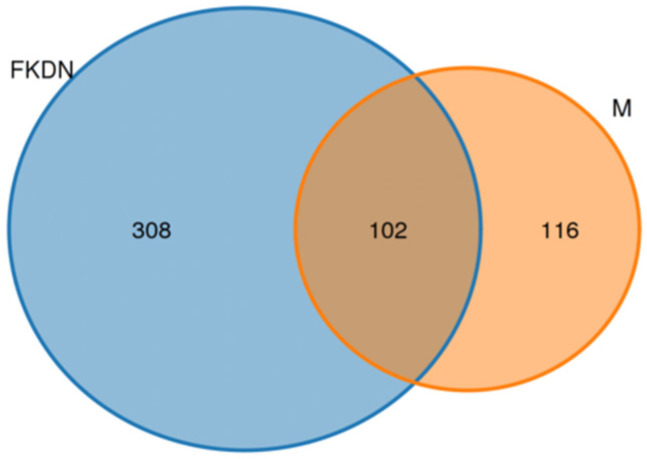
Venn diagram of the characteristics of the two groups of samples. Different colored ellipses represent the number of OTUs annotated by each group, and the size of the ellipses indicates the quantity of the annotated OTUs. The blue ellipse represents the number of OTUs annotated by the Tazhong Town group in the tower, and the orange ellipse represents the number of OTUs annotated by the Pamir Plateau group. The overlapping part in the middle represents the number of OTUs annotated by both groups.

**Figure 3 microorganisms-13-02642-f003:**
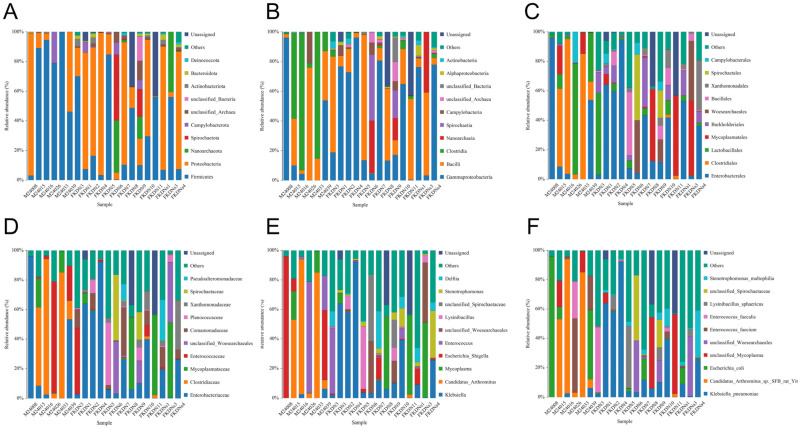
Bar charts showing the distribution of species at various taxonomic levels. The horizontal axis represents the sample names, the vertical axis represents the relative abundance percentage, and the stacked columns represent the top 10 taxonomic groups with the highest relative abundance at each classification level. (**A**) Species distribution bar chart at the phylum level. Different colors represent different phyla. (**B**) Species distribution bar chart at the class level. Different colors represent different classes. (**C**) Species distribution bar chart at the order level. Different colors represent different orders. (**D**) Species distribution bar chart at the family level. Different colors represent different families. (**E**) Species distribution bar chart at the genus level. Different colors represent different genera. (**F**) Species distribution bar chart at the species level. Different colors represent different species.

**Figure 4 microorganisms-13-02642-f004:**
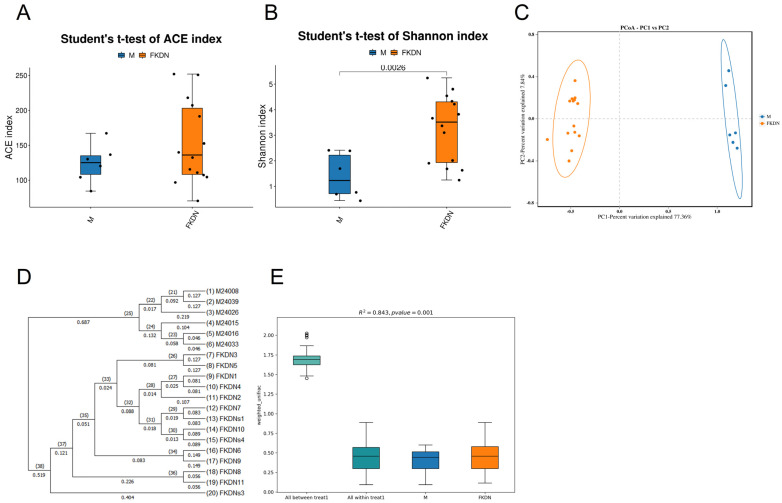
Analysis diagram of Alpha and Beta diversity indices. (**A**) Boxplot of the ACE index (*p* > 0.05). The boxplot shows the within-group distribution of the samples from the two locations. Blue represents the Pamir Plateau group, and orange represents the Tazhong Town group. (**B**) Boxplot of Shannon index (*p* = 0.0026). The boxplot shows the within-group distribution of the samples from the two locations. Blue represents the Pamir Plateau group, and orange represents the Tazhong Town group. (**C**) PCoA analysis diagram. Each point represents a sample; blue indicates the Pamir Plateau group, and orange indicates the Tazhong Town group; the oval circles represent the 95% confidence ellipses. (**D**) UPGMA clustering tree diagram. 1 to 20 represent the sample names, and 21 to 38 represent the branch lengths of the evolutionary branches of the two samples being compared. The closer the samples are, the shorter the branch length, indicating that the species composition of the two samples is more similar. (**E**) PERMANOVA analysis of box plots. The vertical axis represents the Beta distance; the box plot above “All between” represents the Beta distance data of all samples between groups, the box plot above “All within” represents the Beta distance data of all samples within groups, and the subsequent box plots represent the Beta distance data of samples within different groups. The separation situation was evaluated by PERMANOVA (*R*^2^ = 0.843, *p* = 0.001).

**Figure 5 microorganisms-13-02642-f005:**
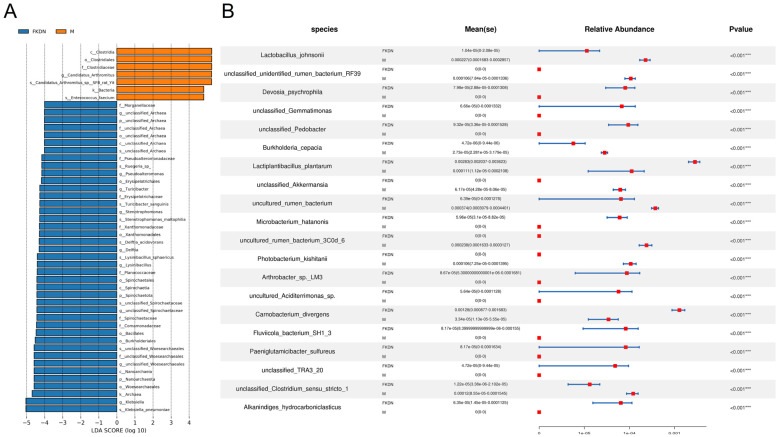
Graph showing significant differences between groups. (**A**) Histogram of LDA values. Blue represents the Tazhong Town group, and orange represents the Pamir Plateau group. The vertical axis shows the classification units that have significant differences between groups. The horizontal axis presents the log scores of LDA analysis for each classification unit in a bar chart format. The classification units are sorted by their score values, and the longer the bar, the more significant the difference in that classification unit. (**B**) Combined graph of Metastats statistical analysis at the species level. The first column shows the species classification information, the second column indicates the group to which the species belongs, the third column presents the mean abundance and standard error, the fourth column shows the relative abundance bar chart of each group, and the *p* value represents the *p*-value of the hypothesis test. Generally, a *p* value less than 0.05 is considered to indicate a significant difference, and *** represents a *p* value less than 0.001. Only the top 20 species are shown in the figure.

**Figure 6 microorganisms-13-02642-f006:**
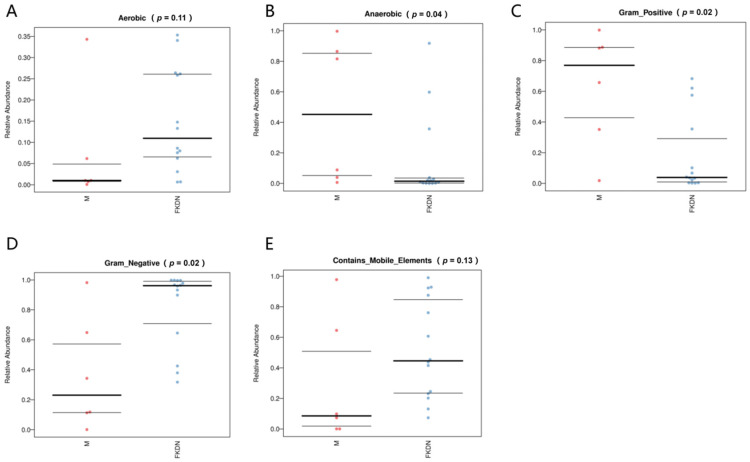
Phenotype prediction diagram of BugBase. (**A**) Relative abundance of aerobic bacteria OTUs; (**B**) Relative abundance of anaerobic bacteria OTUs; (**C**) Relative abundance of Gram-positive bacteria OTUs; (**D**) Relative abundance of Gram-negative bacteria OTUs; (**E**) Relative abundance of bacteria containing mobile genetic elements OTUs. The horizontal axis represents the groups, and the vertical axis represents the relative abundance percentage (displayed as a decimal). The three lines from bottom to top are the lower quartile, the average value, and the upper quartile.

**Table 1 microorganisms-13-02642-t001:** The relative abundances of bacteria with a relative frequency of more than 0.1% at the species level of the two groups.

	Relative Abundance (%)	Pamir Plateau Group (M)	Tazhong Town Group (FKDN)
Dominant Bacteria			
*Candidatus Arthromitus* sp. *SFB rat Yit*		39.76	0.14
*Escherichia coli*		25.26	2.19
*Enterococcus faecium*		12.49	0.45
*Enterococcus faecalis*		4.27	3.52
*Klebsiella pneumoniae*		1.45	23.25
*Lysinibacillus sphaericus*		0	4.80
*Stenotrophomonas maltophilia*		1.58 × 10^−3^	4.57

The leftmost column represents the bacterial species with a relative abundance of >0.1% in the two groups of saxaul sparrow fecal samples. The middle column represents the relative abundance of each bacterial species in the Pamir Plateau group (in percentage). The rightmost column represents the relative abundance of each bacterial species in the Tazhong Town group (in percentage).

## Data Availability

The original contributions presented in this study are included in the article/Appendix A. Further inquiries can be directed to the corresponding authors.

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
