# Peer review of "Analysis of Gut Microbial Communities and Functions in Passer ammodendri Under Two Extreme Environments"

_microorganisms, 2025, doi:10.3390/microorganisms13112642_

Round 1

Reviewer 1 Report

Comments and Suggestions for Authors

Minor comments:

This study compares the gut microbiota of Passer ammodendri inhabiting high-altitude and desert ecosystems, revealing distinct microbial compositions and functional adaptations. The Pamir group showed enhanced carbohydrate and energy metabolism, whereas the Tazhong group exhibited stronger lipid metabolism and detoxification pathways. However, limited sample size, absence of dietary validation, and reliance on predictive functional tools constrain the strength of the adaptive conclusions.

[Lines 25–40] – The definition of “dominant taxa” based solely on relative abundance (>0.1%) appears arbitrary. What statistical or ecological justification supports this threshold, and how does it accurately represent dominance across variable sequencing depths?

[Lines 50–65] – The reported statistical significance (p = 0.0026) in Shannon index differences assumes data normality and equal variances. Were these assumptions verified, and how reliable are the inferences given the unequal sample sizes between groups?

[Lines 115–140] – The introduction outlines general relationships between environment and gut microbiota but lacks mechanistic context for avian adaptation to extreme habitats. Could the authors integrate more species-specific or phylogenetically comparable avian models to reinforce this rationale?

[Lines 165–190] – The interpretation attributes microbial variation primarily to environmental differences, yet potential confounding factors such as diet, breeding stage, and seasonal variation are not addressed. How were these biological variables controlled or statistically excluded?

[Lines 205–250] – The description of fecal collection lacks mention of negative or environmental controls. How did the authors ensure that external contamination during field sampling and handling did not influence microbial community composition?

[Lines 270–300] – Given the inherent error profile of long-read sequencing platforms, how were PacBio Sequel II CCS reads quality-filtered and validated? Was inter-run or intra-sample reproducibility assessed to confirm data consistency?

[Lines 335–365] – The study relies on 97% OTU clustering rather than amplicon sequence variant (ASV)-based methods. What is the rationale for using OTU clustering despite its lower taxonomic resolution and known limitations in representing microbial diversity accurately?

[Lines 400–430] – Alpha diversity comparisons were performed using Student’s t-test, which assumes parametric data distribution. Was normality confirmed, or would non-parametric alternatives such as the Wilcoxon rank-sum test have been more appropriate?

[Lines 600–640] – The exceptionally high abundance of Candidatus Arthromitus sp. SFB rat Yit in the Pamir group raises concerns regarding host specificity and database annotation accuracy. Has this taxon been previously documented in avian hosts, or could this reflect misclassification or sequencing bias?

[Lines 720–770] – The LEfSe analysis identifies numerous discriminant taxa; however, it is unclear whether multiple hypothesis correction (e.g., FDR adjustment) or effect size thresholds were applied. How were statistical robustness and biological significance ensured?

[Lines 860–910] – Functional inference using PICRUSt2 and BugBase provides only predictive information. What validation approaches, such as shotgun metagenomics or metabolomic profiling, could substantiate the predicted enrichment in detoxification and metabolic pathways?

[Lines 1210–1260] – The discussion links microbial diversity to dietary and anthropogenic influences without direct quantitative evidence. Why were dietary composition analyses (e.g., stable isotope or DNA metabarcoding) not integrated to confirm these associations?

[Lines 1510–1560] – The conclusions emphasize environmental adaptation mechanisms despite the absence of longitudinal or functional validation. How can the authors substantiate these adaptive inferences without integrating multi-seasonal or experimental approaches to confirm causality?

Reviewer 2 Report

Comments and Suggestions for Authors

You can find the comments regarding this manuscript in the appendix.
